# Genomic Steppe ancestry in skeletons from the Neolithic Single Grave Culture in Denmark

**Anne Friis-Holm Egfjord[1], Ashot Margaryan[1,2], Anders Fischer[3,4], Karl-Göran Sjögren[3], T. Douglas Price[5], Niels N. Johannsen[6], Poul Otto Nielsen[7], Lasse Sørensen[7], Eske Willerslev[1,8], Rune Iversen[9], Martin Sikora[1], Kristian Kristiansen[3]\*, Morten E. Allentoft[1,10]\***

**1** Lundbeck Foundation GeoGenetics Centre, GLOBE Institute, University of Copenhagen, Copenhagen, Denmark, **2** Center for Evolutionary Hologenomics, University of Copenhagen, Copenhagen, Denmark, **3** Department of Historical Studies, University of Gothenburg, Gothenburg, Sweden, **4** Sealand Archaeology, Kalundborg, Denmark, **5** Laboratory for Archaeological Chemistry, University of Wisconsin, Madison, Wisconsin, United States of America, **6** Department of Archaeology and Heritage Studies, Aarhus University, Højbjerg, Denmark, **7** Ancient Cultures of Denmark and the Mediterranean, The National Museum of Denmark, Copenhagen K, Denmark, **8** Department of Zoology, University of Cambridge, Cambridge, United Kingdom, **9** Saxo Institute, Section of Archaeology, University of Copenhagen, Copenhagen, Denmark, **10** Trace and Environmental DNA (TrEnD) Laboratory, School of Molecular and Life Sciences, Curtin University, Perth, Australia

\* kristian.kristiansen@archaeology.gu.se (KK); morten.allentoft@curtin.edu.au (MEA)

**Data Availability Statement:** Sequence data are available at the European Nucleotide Archive under accession number PRJEB40059.

## Abstract

The Gjerrild burial provides the largest and best-preserved assemblage of human skeletal material presently known from the Single Grave Culture (SGC) in Denmark. For generations it has been debated among archaeologists if the appearance of this archaeological complex represents a continuation of the previous Neolithic communities, or was facilitated by incoming migrants. We sampled and analysed five skeletons from the Gjerrild cist, buried over a period of c. 300 years, 2600/2500–2200 cal BCE. Despite poor DNA preservation, we managed to sequence the genome (>1X) of one individual and the partial genomes (0.007X and 0.02X) of another two individuals. Our genetic data document a female (Gjerrild 1) and two males (Gjerrild 5 + 8), harbouring typical Neolithic K2a and HV0 mtDNA haplogroups, but also a rare basal variant of the R1b1 Y-chromosomal haplogroup. Genome-wide analyses demonstrate that these people had a significant Yamnaya-derived (i.e. steppe) ancestry component and a close genetic resemblance to the Corded Ware (and related) groups that were present in large parts of Northern and Central Europe at the time. Assuming that the Gjerrild skeletons are genetically representative of the population of the SGC in broader terms, the transition from the local Neolithic Funnel Beaker Culture (TRB) to SGC is not characterized by demographic continuity. Rather, the emergence of SGC in Denmark was part of the Late Neolithic and Early Bronze Age population expansion that swept across the European continent in the 3rd millennium BCE, resulting in various degrees of genetic replacement and admixture processes with previous Neolithic populations.

**Funding:** KK received funding by the Swedish Riksbanken (The Swedish Foundation for Humanities and Social Sciences: https://www.riksbank.se/en-gb/) grant M16-0455:1 - Towards a New Prehistory. MEA is funded by the Independent Research Fund Denmark (Sapere Aude, grant 7027-00147B). Centre for GeoGenetics is funded by the Lundbeck Foundation. The funders had no role in study design, data collection and analysis, decision to publish, or preparation of the manuscript.

**Competing interests:** The authors have declared that no competing interests exist.

# Introduction

Recent large-scale ancient population genetic studies have shown that the Corded Ware Culture (CWC) in Europe emerged as groups connected to the Yamnaya Culture who expanded westwards from the Pontic-Caspian steppe and admixed with European Neolithic populations during the 3rd millennium BCE [1–5]. This migration process changed forever the cultural and genetic landscape of Europe. The appearance of the Single Grave Culture (SGC) in Denmark, Northern Germany and Netherlands c. 4850–4600 years ago represents a cultural shift similar to the formation of the CWC in other parts of Europe, and for decades this has been a widely discussed phenomenon among archaeologists. In Denmark, the initial SGC settlement area was the flat sandy soils of central and western Jutland where no skeletal remains are preserved, but thousands of small barrows testify to a massive presence of this culture [6]. Its arrival was accompanied by changes to the landscape as documented in pollen analysis [7–11].

In Jutland (western Denmark), the SGC is primarily known from burials. Settlement evidence is sparse, and the remains of buildings derive almost exclusively from the later phases of the culture. The graves are normally single inhumations under small mounds. Soil marks reflecting the position of bodies in the graves show that they were placed on their side in contracted (hocker) position, generally on the east-west axis and facing south. Bodies lying on their right side (head to the west) are usually accompanied by battle axes, flint axes and amber discs, while bodies lying on the left side (head to the east) have pottery and amber beads, suggesting a burial treatment emphasising sex divisions, in line with the overall CWC tradition [12]. Evidence of subsistence is poor but cultivation of emmer and naked barley is attested, as well as keeping of livestock [13–15]. Pollen evidence from western Jutland shows widespread clearings at this time, primarily in the form of pastures [8–10] and the purpose was clearly to create large tracts of open land for grazing [11]. In eastern Denmark, SGC presence is both later and less prominent and it takes different forms. Here, the classical SGC mounds do not occur, with a few exceptions, instead existing megalithic tombs are reused, and SGC traditions are merged with older Funnel Beaker Culture (TRB) traditions [16]. Also, new types of megalithic tombs are built in northern Denmark, such as the Bøstrup type cists, of which the Gjerrild burial monument (Fig 1) is an example.

It has long remained an unresolved question whether the development of the SGC in Denmark was facilitated by migrations of people (from southern areas) or represents a local cultural adaptation and settlement expansion by indigenous Neolithic farmers of the TRB, or a combination of the two phenomena [7, 17, 18]. The discussion is complicated by the presence of the coastal, mixed-economy connected to the Pitted Ware Culture (PWC) along the Danish coasts around the Kattegat Sea, which to a larger extent relied on non-domestic resources, including fish and seal. The latter archaeological complex overlaps chronologically with the early SGC and has a strong presence in northern Djursland where the Gjerrild monument is located [19, 20].

Despite the presence of thousands of SGC barrows in Jutland, Northwestern Germany and Netherlands, human remains are very sparse, due to the prevailing soil conditions in the areas where SGC groups preferred to settle. In this respect, the relatively well-preserved skeletons from the Gjerrild burial (Fig 1) are fortunate exceptions. To investigate if the SGC individuals from Gjerrild were local people of typical Neolithic genetic ancestry, who simply adopted a new burial culture, or if they represent immigrant CWC groups, we sampled five of the Gjerrild skeletons for ancient DNA (aDNA) analyses, $^{14}$C-dating and $^{13}$C/$^{15}$N dietary measurements of collagen. We employed Next Generation 'shotgun' sequencing to obtain genome-wide information, allowing us to compare their ancestry with other relevant Neolithic and Early Bronze Age groups of Europe. Our study provides detailed insight into the individuals

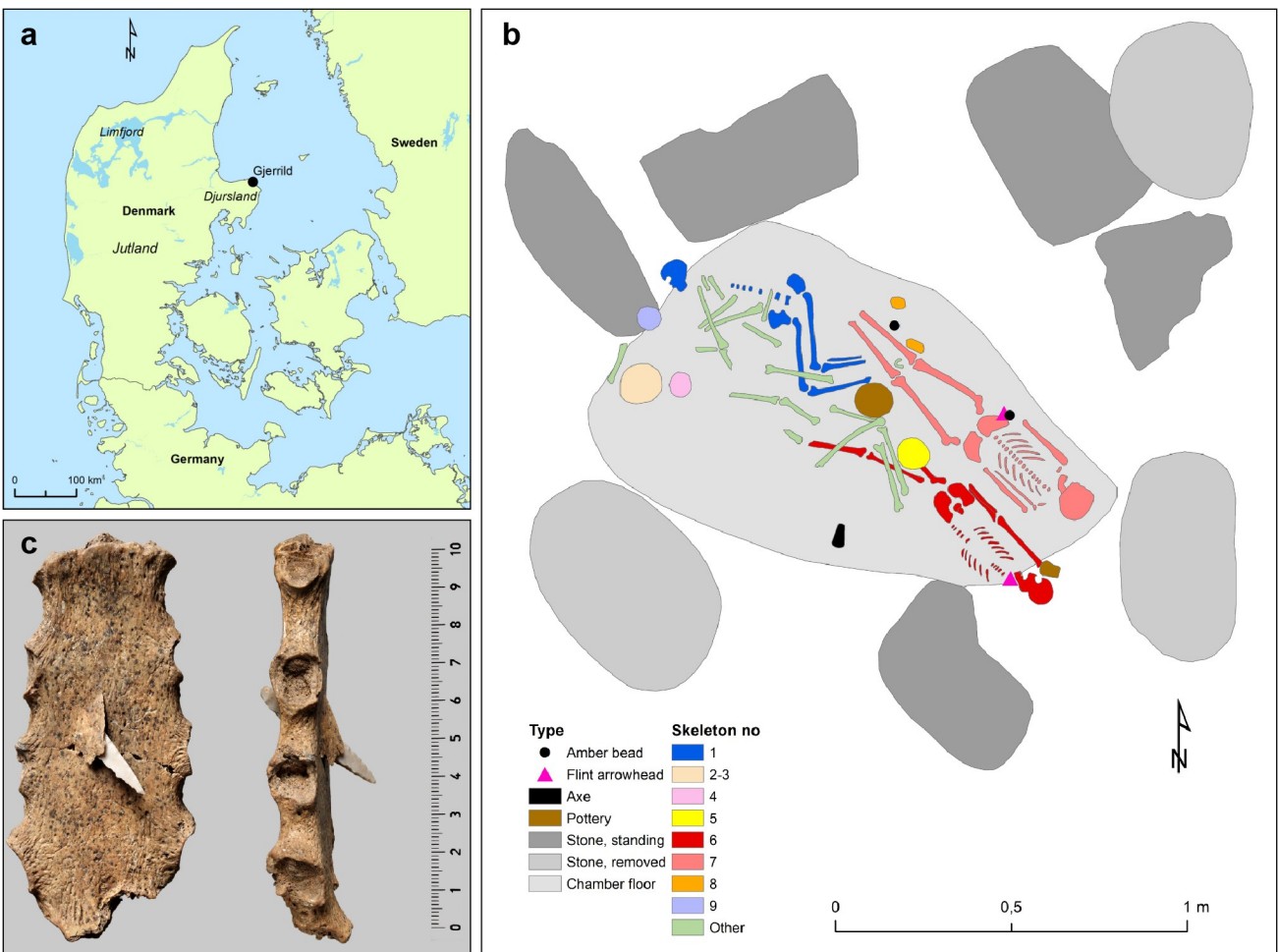

**Fig 1. A general overview of the Gjerrild burial monument.** A) Map of Denmark with the location of Gjerrild, based on public domain data from Natural Earth (https://www.naturalearthdata.com). B) plan of the Gjerrild chamber tomb with identified individuals. Redrawn by Karl-Göran Sjögren from an excavation plan by C. L. Vebæk. Individual 1: female, 20–25 years old, placed in hocker position. Individuals 2–4: scattered fragments of 3–4 individuals. Individual 5: male child, c. 2 years old. Individual 6: male, 20–30 years, with type D arrowhead in chest bone. Individual 7: male, 40–50 years, with trepanated cranium and with atypical tanged arrowhead plus amber bead by the right hip. Individual 8: male child, 5–6 years old. Individual 9: cranium of male, 40–50 years. C) breast bone of individual 6, fatally penetrated by a flint arrow head. Photo: John Lee, Danish National Museum, courtesy Samantha Reiter.

buried in this exceptional Danish prehistoric site, and contributes to understanding the role of migration and the formation of new cultures during the 3rd millennium in Northern Europe.

## The Gjerrild site

The Gjerrild stone cist in northern Djursland, eastern Jutland (Fig 1), is remarkable for containing the largest and best-preserved assemblage of SGC skeletons known from Denmark. From this follows a unique opportunity to obtain information on the genetic ancestry of people representing the SGC in Denmark. In the cultural history of Neolithic Denmark, northern Djursland is peculiar as this area lacks finds from the final TRB period but instead experienced a relatively short but significant hunter-fisher-gatherer oriented PWC phase (c. 3000–2700 cal BCE). After the PWC phase, the cultural development on Djursland followed that of eastern Denmark, characterized by the continued use of megalithic tombs, few SGC single graves and few and late battle axes [21]. Thus, only four SGC graves have been recorded in the former

PWC-dominated northern part of Djursland and these are all more recent in date, i.e. after 2500 cal. BCE (cf. [6], Figs 470–476, catalogue nos. 543–546).

The Gjerrild grave was excavated in 1956 and the burial was archaeologically dated to the SGC's Upper Grave period (c. 2450–2250 cal. BCE) based on the grave goods, which consisted of a thin-bladed thick-butted flint adze, two pottery vessels, two amber beads and three tanged flint arrowheads (type D). Unlike most of the classical SGC graves, the Gjerrild barrow contained a megalithic chamber, a so-called Bøstrup cist. These are large stone cists dated to the later part of the SGC (Ground and Upper Grave period, i.e. after 2600 cal. BCE) constructed with an entrance, which allowed for succeeding burials. The Bøstrup cists are almost exclusively found in northern and north-eastern Jutland and thus represent a later expansion of the SGC, north and east of the early core areas of central and western Jutland. The chamber had been disturbed but the outline could be reconstructed. It was 2.8–3 m long and 1.6–1.7 m wide in the broader northern end, while it was only c. 1m wide in the south, where the entrance was found [22, 23]. Osteological analyses were carried out by Jørgen Balslev Jørgensen and by Pia Bennike [23, 24], and the teeth were analysed by Verner Alexandersen [25]. Hans Christian Petersen carried out a comparative osteological analysis of two Gjerrild craniums that were compared to skulls of the previous TRB, as well as to those of CWC individuals from central Germany, and from Danish Late Neolithic individuals [26]. Based on this admittedly small sample, it was concluded that the Gjerrild crania grouped with representatives of central German CWC, while individuals from the succeeding Late Neolithic period in Denmark were a mixture of both TRB and CWC groups [26].

The grave contained at least ten individuals: six adults, one juvenile and three children. Several of the skeletons were articulated, while others were more or less disturbed and disarticulated. Balslev Jørgensen identified remains from several individuals, numbered 1–9 (Fig 1). Two skeletons (Gjerrild 6 and 7) were lying parallel to each other in extended supine positions, while two others were in hocker position. A striking feature of Gjerrild individual 6 is a tanged flint arrowhead lodged in the breastbone (Fig 1), probably a battle wound that caused his death. Type D arrowheads are thought to characterize the later SGC phase (c. 2525–2250 cal. BCE, [6]: 438–9) but our radiocarbon date of the Gjerrild 6 skeleton, presented herein is the first "direct" dating of such an arrowhead. A notable feature about Gjerrild 7 was a hole in the skull from a trepanation, probably to treat health problems such as releasing blood build-up from a traumatic injury [24]: 76–77.

## Materials and methods

### The samples

Five human skeletons from the Gjerrild chamber were sampled for this study (Gjerrild 1, 5, 6, 7, 8) and the material consisted of three teeth, one petrous bone and one fibula bone (Table 1

**Table 1. Sample information, $^{14}$C dates and isotope measurements.**

| Skeleton # | Sample type | Age class | Sex | Uncal. BP | Cal. BCE (95.4%) | Cal. BCE rescorr (95.4%) | Cultural period | δ$^{13}$C ‰VPDB | δ$^{15}$N ‰AIR |
|---|---|---|---|---|---|---|---|---|---|
| Gjerrild 1 | Tooth | Adult | Female | 3950±31 | 2571–2310 | 2569–2306 | SGC | -20.8 | 8.3 |
| Gjerrild 5 | Petrous Bone | Child | NA | 3790±34 | 2343–2055 | 2284–2035 | SGC/LN | -19.3 | 11.5 |
| Gjerrild 6 | Tooth | Adult | Male | 3875±29 | 2474–2298 | 2466–2287 | SGC | -19.7 | 9.4 |
| Gjerrild 7 | Fibula Bone | Adult | Male | 3877±21 | 2462–2287 | 2447–2201 | SGC | -19.3 | 9.4 |
| Gjerrild 8 | Tooth | Child | NA | 4007±36 | 2624–2462 | 2575–2345 | SGC | -19.4 | 12.4 |
| Mandible | Tooth | NA | NA | 3410±26 | 1866–1622 | 1741–1443 | EBA | -19.2 | 8.7 |

Osteologically determined age class of death, osteologically determined sex, radiocarbon age, calibrated age ranges (95.4%) and isotope results.

and S1 Table). Additionally, we sampled a disarticulated mandible that was only used for AMS dating and $\delta^{13}C$ and $\delta^{15}N$ analyses. The Gjerrild cist was excavated in 1955 in Gjerrild, Eastern Jutland, locality no. 140106–9, by the Danish National Museum of Denmark, where the remains are kept (NM inv. nos. A44572-78a). All necessary permits were obtained from the National Museum of Denmark for the described study, which complied with all relevant regulations.

## Radiocarbon dating and isotope measurements

The radiocarbon measurements were performed in the Oxford and Belfast AMS-dating laboratories, using their standard laboratory procedures, reported in [27] and [28]. All samples had collagen quality measures fulfilling the requirements and were dated successfully. One individual, Gjerrild 7, was dated twice at the Oxford laboratory as part of their standard quality assessment procedures. A combined date has been calculated for this individual. Calibrations were done using by OxCal 4.3.2 [29] and the IntCal13 calibration curve. Collagen $\delta^{13}C$ and $\delta^{15}N$ values were measured separately on dedicated mass spectrometers.

## DNA extraction, library preparation and sequencing

All the pre-amplification laboratory work was performed in clean laboratory facilities at the Lundbeck Foundation GeoGenetics Centre (Globe Institute, University of Copenhagen), according to strict aDNA guidelines [30, 31]. The outer surfaces of the samples were removed using a sterile cutting disc to reduce DNA contamination. For teeth, the crown was initially separated from the root by a cutting disc and saved for radiocarbon- and strontium isotope analyses [32]. The inner dentine was then removed with a pointy drilling bit, thereby targeting the outer cementum layer for DNA extraction, in order to maximize the endogenous DNA yield [33]. The petrous bone was cut into slices by a cutting disc until the dense otic capsule was reached. The fibula bone was sampled with a cutting disc and the densest part of the bone was targeted, avoiding the spongy parts [34]. The samples were crushed into smaller pieces with a pincher and ranged in mass from 56 to 707 mg. To increase the endogenous DNA yield, we performed a brief 'pre-digestion' step prior to the extraction protocol [33]. The digestion buffer contained pr. ml 929 uL 0.5M EDTA, 10 uL TE buffer, 10 uL Proteinase K, 50 uL 10% N-laurylsarcosine, 1 uL phenol red. A volume of 2 ml was added to each crushed sample, which was then vortexed and incubated with rotation for 15 min at 42˚C, after which the samples were spun down, and the supernatant discarded. Following this pre-digestion, we added 3.5mL of fresh digestion buffer to each sample and incubated them for 24h under the same conditions as above. The DNA was then purified using the silica-in-solution method similar to Rohland & Hofreiter (2007) [35], but using the optimized binding buffer from [1]. Double-stranded blunt-end libraries were constructed from the extracted DNA using NEBNext DNA Prep Master Mix Set E6070 (New England Biolabs Inc.) with protocol modifications [1, 36], and amplified with indexed Illumina-specific adapters prepared as in [37]. The DNA concentration of each amplified library was quantified on an Agilent 2200 Tapestation, and for the initial shotgun screening run, the libraries from all five Gjerrild samples were pooled equimolarly. Sequencing (80bp, single read) for this project was performed on Illumina HiSeq 2500 or Illumina HiSeq 4000 platforms at the Danish National High-throughput DNA Sequencing Centre.

## Data processing

The raw sequencing data were base-called using the Illumina software CASAVA 1.8.2 and sequences were de-multiplexed with a requirement of full match of the six-nucleotide index

that was used during library preparation. The reads were trimmed using AdapterRemoval v.2.2.2 [38] by removing adapter sequences and stretches of ambiguous bases, retaining only trimmed sequences with a minimum length of 30 bp. The trimmed reads were mapped to the human reference genome build 37.1 using BWA v.0.7.15 aln and samse [39]. Due to the overall short read length, the seeding parameter was disabled, requiring a more sensitive alignment [40, 41]. Only aligned reads with mapping quality of at least 30 were kept and sorted by chromosome position using SAMtools v.1.8.11 [39]. Final data was merged to library level and sequence duplicates were removed using Picard v.2.9.4 MarkDuplicates (http://broadinstitute. github.io/picard/). Finally, the data were merged to sample level and sequences were locally realigned around indels using GATK v.3.7 RealignerTargetCreator and IndelRealigner [42], MD-tags were provided using SAMtools calmd and the data were indexed using SAMtools index. The average depth of coverage (DoC) on the nuclear and mitochondrial genomes was calculated for each sample using Genobox (https://github.com/srcbs/GenoBox). Statistics of the read data processing are shown in S2 Table.

**Screening and sample selection.** To evaluate DNA preservation and potential for obtaining genome-wide data, an initial 'screening' sequencing run was performed with all five Gjerrild libraries pooled together. The criteria for selecting candidates for deeper sequencing was inspired by previous guidelines [1], excluding samples with an endogenous human DNA content <0.5% and <10% C-T damage in the 5'ends (see below). Furthermore, library clonality and overall sequencing efficiency was evaluated. Libraries from the three best samples (Gjerrild 1,5 and 8) were selected for deeper sequencing with one library sequenced per lane, and the data were merged to sample level as outlined above.

**Ancient DNA authentication, genetic sex and haplogroups.** To confirm the ancient origin of the DNA molecules, post mortem DNA damage patterns were determined using mapDamage2.0 [43]. Cytosine deamination damage was recorded as C to T transitions toward the 5' end of the DNA reads. Further, DNA contamination was estimated from both mtDNA (both sexes) and the X-chromosome (only males). For samples having mitochondrial genomes with >5X depth of coverage, contamination levels were determined using the *contamMix* software [44] using only sites with a minimum depth of coverage of 3X, and 7 bp from each end of the mtDNA reads were disregarded to minimize the biases introduced by DNA damage. For males with an X-chromosome coverage >0.5X, contamination was estimated using ANGSD [45]. Pseudoautosomal regions were removed for the analysis and mapping quality ≥30 and base quality ≥20 filters were used. The sex of the Gjerrild individuals was determined genetically by using the Ry parameter from [46]. The Y-chromosome haplogroups were inferred by determining ancestral / derived status for haplogroup-determining SNPs from the International Society of Genetic Genealogy (ISOGG, http://www.isogg.org, version 14.255). Phylogenetic placement of the Gjerrild 5 Y-chromosome (showing higher genomic coverage than the other samples) was carried out using the evolutionary placement algorithm, implemented in the EPA-ng package [47]. The reference tree was built from the 1000 Genomes project Y-chromosome SNP calls [48] (ftp://ftp. 1000genomes.ebi.ac.uk/vol1/ftp/release/20130502/supporting/chrY/ALL.chrY_10Mbp_mask. glia_freebayes_maxLikGT_siteQC_phyloImputedV5.20130502.60555_biallelic_snps.vcf.gz) using raxml-ng [49] with the GTR+G model and ascertainment correction ("—model GTR+G+ASC_LEWIS" option). The mitochondrial DNA (mtDNA) haplogroups were determined using HaploGrep2 [50].

**Population genomics.** Genetic clustering of the Gjerrild individuals was carried out using a merge with a reference dataset of 297 previously published ancient Eurasians with whole-genome shotgun sequencing data (See S3 Table for full list and references). Pseudohaploid genotypes for ancient individuals were obtained by randomly sampling a high-quality allele (mapping quality ≥ 30, base quality ≥ 30) from sequencing reads overlapping 12,731,663

biallelic transversion SNPs from the 1000 Genomes project data. We carried out genetic clustering of the ancient individuals by multidimensional scaling (MDS), using pairwise genetic distances calculated as 1- p(IBS), where p(IBS) is the genome-wide fraction of alleles shared identical-by-state (IBS) between individuals. Model-based clustering was carried out using ADMIXTURE [51], for K = 2 to K = 6 clusters, and 10 replicates for each value of K. Major modes across replicates were selected and aligned across runs using pong [52].

## Results

### Isotope measurements and radiocarbon dating

Collagen $\delta^{13}C$ and $\delta^{15}N$ were measured for the purpose of estimating the protein sources in the diet, and for estimating marine reservoir effect correction on the radiocarbon dates. Both isotopes show low values, indicating limited contributions from marine and/or freshwater sources, and thus limited or no reservoir effects (Table 1, S1 Table and Fig 2). The results from Gjerrild 1 indicate a completely terrestrial diet, while values from Gjerrild 6, 7, and the loose mandible has a limited marine signal. A rough estimate suggests a possible marine reservoir effect for these three individuals at c. 45–60 years, supposing a marine reservoir effect at 273 ±18 years for a fully marine diet [53]. Gjerrild 5 and 8 represent children affected by lactation

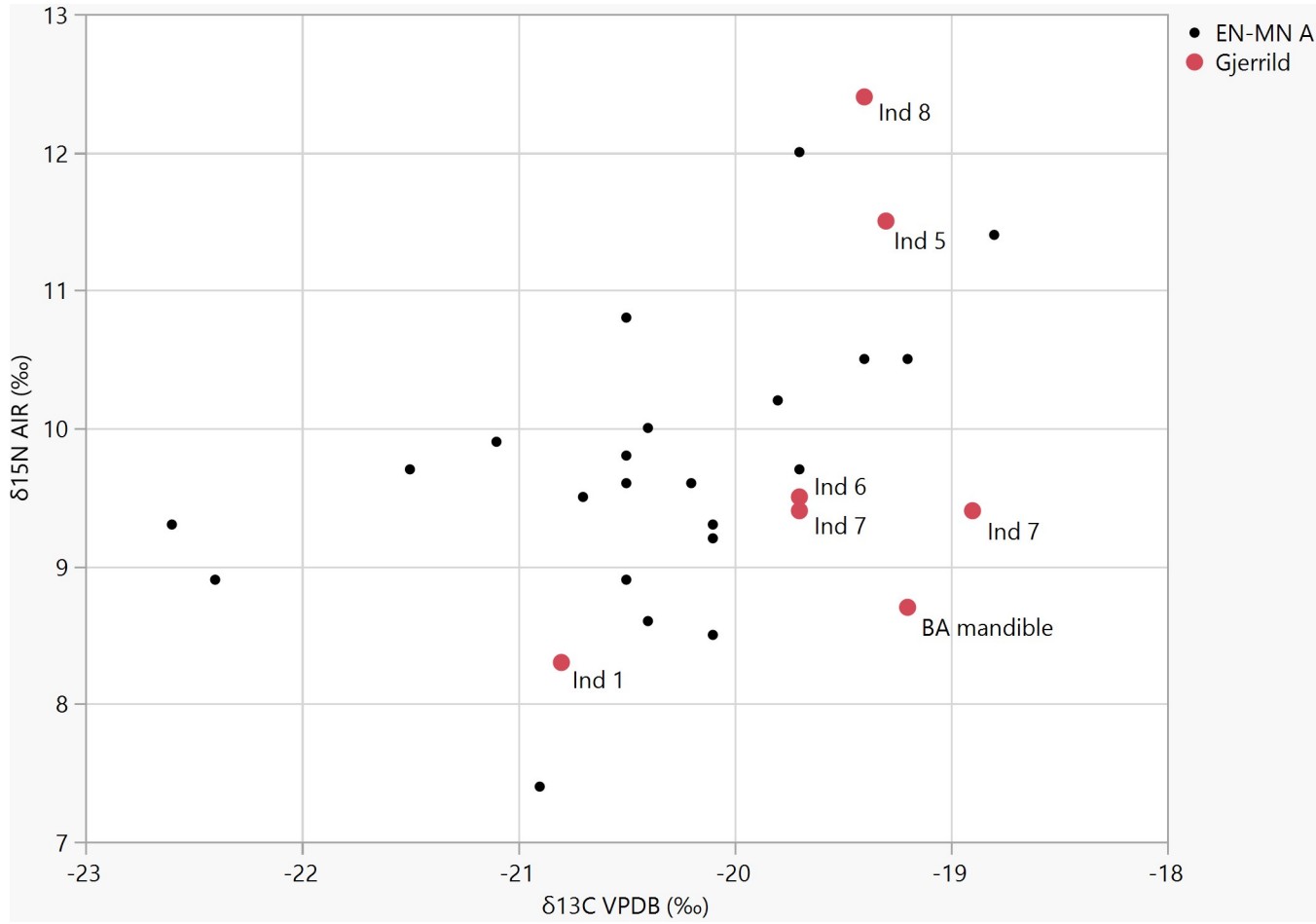

**Fig 2. Isotope measurements.** $\delta^{13}C$ and $\delta^{15}N$ values from Gjerrild compared to values from other Early (EN) and Middle (MN A) Neolithic Danish individuals attributed to the TRB culture (data from Fischer et al. 2007 [69]). Numbers denote the different skeletons from Gjerrild. Individuals 5 and 8 are children (Inf I).

but should also have minimal reservoir effects (see discussion). Full details of $^{14}$C and isotope measurements are provided in S1 Table.

Five of the six radiocarbon dates span 4007–3790 uncal BP, corresponding to 2575–2035 cal BCE after reservoir correction (Table 1, S1 Fig and Fig 2). Gjerrild 8 is the oldest of these skeletons, calibrated to 2575–2345 cal BCE (95.4%), whereas Gjerrild 5 is the youngest, calibrated to 2284–2035 cal BCE (95.4%). The dates suggest a main period of use from c. 2600/2500 to around 2200 cal BCE, confirming that the chamber was indeed used during the SGC epoch. We can now conclude that individuals 1, 6, 7 and 8 belong to the Ground and Upper grave periods, i.e. the middle and late parts of the SGC. This implies an earlier monument construction and a longer use period than the archaeological typology had indicated. The slightly younger date of individual 5 corresponds to the Late Neolithic in the chronology of [54], or to the transition Middle/Late Neolithic in the chronology of [55]. The younger date for the loose mandible (Table 1) corresponds to the Early Bronze Age. The two males Gjerrild 6 and 7 have overlapping dates and may well be contemporary, while Gjerrild 8 is most likely earlier and Gjerrild 5 later. The date of Gjerrild 1 largely overlaps with Gjerrild 8 but could also be slightly later, i.e. intermediate between Gjerrild 8 and 6/7.

## DNA preservation, authenticity and genomic coverage

Initial shotgun sequencing revealed poor DNA preservation with low endogenous DNA content, ranging from 0.1–2.6%, except from Gjerrild 5 yielding 14.7% (Table 2), underlining that petrous bones will often outperform other skeletal substrates for aDNA analysis [56, 57]. The low endogenous DNA content made it difficult to obtain sufficient genome wide information for several of the samples. The subsequent sequencing effort was therefore focused on Gjerrild 1, 5 and 8, resulting in genomic depth of coverage (DoC) values of 0.007X, 1.038X and 0.020X, respectively (Table 2). All three samples showed characteristic signs of post mortem DNA damage, with increased frequency of C to T misincorporations ranging from 18.9–29.8% at the first 5' position (Table 2). Moreover, the average lengths of the sequenced human DNA fragment were very short, ranging from 39–45.5 base pairs (bp). The mtDNA-based contamination estimates proved low, ranging from 0.09 to 0.73% (Table 2). The X-chromosome based contamination estimate could only be performed for Gjerrild 5 (being a male with >0.5X DoC), and again showing a negligible contamination level (2.5%). In summary, these DNA damage and contamination results testify to the authenticity of the DNA in all three samples. Full details of the sequencing results are provided in S2 Table.

## Genetic sex determination and haplogroups

Gjerrild 5 and 8 were genetically identified as males and Gjerrild 1 as a female (Table 2). The latter had been determined previously on osteological grounds, whereas Gjerrild 5 and 8 could

**Table 2. Basic sequencing results.**

| | | | | | | Genetix sex | Hapologroups | |
|---|---|---|---|---|---|---|---|---|
| Sample | Endo% | DoC | MT DoC | 5' C-T % | MT contam. % | Ry (CI) | mtDNA | Y |
| Gjerrild 1 | 0.6 | 0.007 | 5.9 | 25.1 | 0.23 (0.04–5.23) | XX, 0.004 (0.004–0.005) | HV0 | NA |
| Gjerrild 5 | 14.7 | 1.04 | 52.8 | 29.8 | 0.73 (0.11–2.03) | XY, 0.081 (0.081–0.082) | K2a | R1b1 |
| Gjerrild 8 | 0.9 | 0.02 | 21.7 | 18.9 | 0.09 (0.02–1.94) | XY, 0.085 (0.082–0.088) | K2a | NA |

*Endo%* is the endogenous DNA content, being the proportion of sequences identified as human. *DoC* and *MT DoC* is the genomic depth of coverage of the genome and mitochondrial genome, respectively. *5'C-T %* is the proportion of C to T misincorporation damage at the first position of the 5' end. *Genetic sex* was estimated with the Ry value [46] shown here with 95% CI. *MT Contam %* is the contamination proportion (and 95% confidence interval) based on mtDNA. *mtDNA and Y-chromosome haplogroups* are also shown.

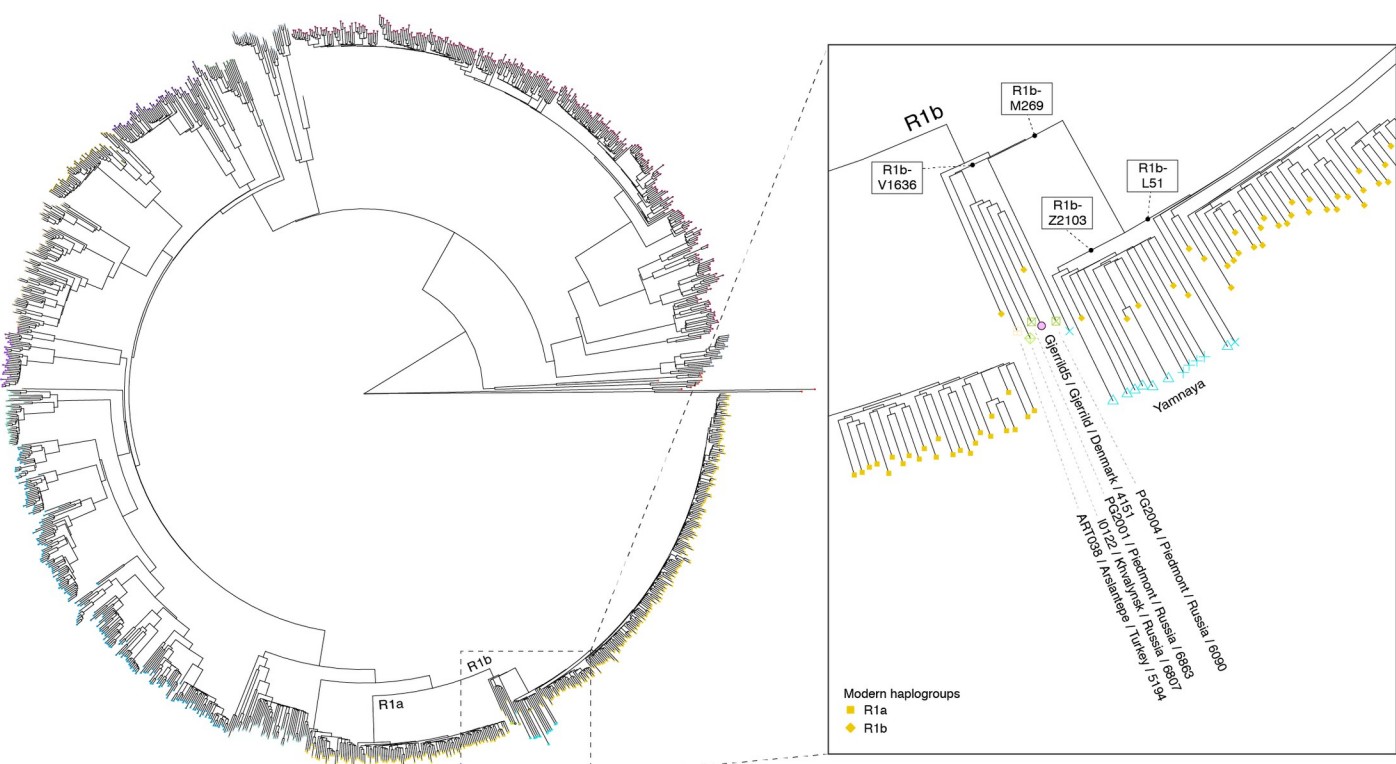

**Fig 3. Phylogenetic placement of the Gjerrild 5 Y-chromosome haplogroup.** Phylogenetic placement of Y chromosome sequences of Gjerrild 5 and selected ancient individuals with R1b haplogroups. Terminal branch lengths of ancient individuals in inset are not drawn to scale, to aid visualization of placements.

not be assigned to sex based on osteology (Table 1) and are thus identified for the first time in this study. The mitochondrial haplogroup K2a was assigned for both Gjerrild 8 and 5 with high confidence (99%), while Gjerrild 1 was assigned HV0 (92% confidence) (Table 2). The Y-chromosome haplogroup of Gjerrild 5 was determined as R1b-V1636 (R1b1a2), a rare sub-clade of R1b (S4 Table). Phylogenetic placement of the Y chromosome sequence of Gjerrild 5 together with ancient individuals previously found to carry R1b1a2 [2, 58, 59] shows that the divergence of R1b-V1636 pre-dates the expansion and diversification of the R1b-M269 lineage. Sub-haplogroups of R1b-M269 include R1b-Z2103, which is common among ancient individuals associated with Yamnaya burials, as well as R1b-L51, common across present-day Europeans (Fig 3).

## Population genomics

Pseudohaploid genotyping of the Gjerrild individuals resulted in a total of 101,541 SNPs (0.8%) covered for Gjerrild 1, 8,928,427 SNPs (70.1%) for Gjerrild 5 and 298,837 SNPs (2.4%) for Gjerrild 8, out of a total of 12,731,663 transversion SNPs in the 1000 Genomes reference panel Multidimensional scaling (MDS) on a distance matrix derived from pairwise allele-sharing of ancient individuals, as well as projection of ancient samples on a principal component analysis from present-day west Eurasian populations, showed the already well-established clustering of the ancient individuals related to cultural/$^{14}$C date, geography and cultural contexts [1, 2, 60–62] (Fig 4 and S2 Fig). The first dimension in the MDS plot separates individuals of Neolithic contexts from Central Asian Bronze Age individuals (i.e. Okunevo, Botai), with individuals from European and Steppe Bronze Age contexts aligned on a cline between them.

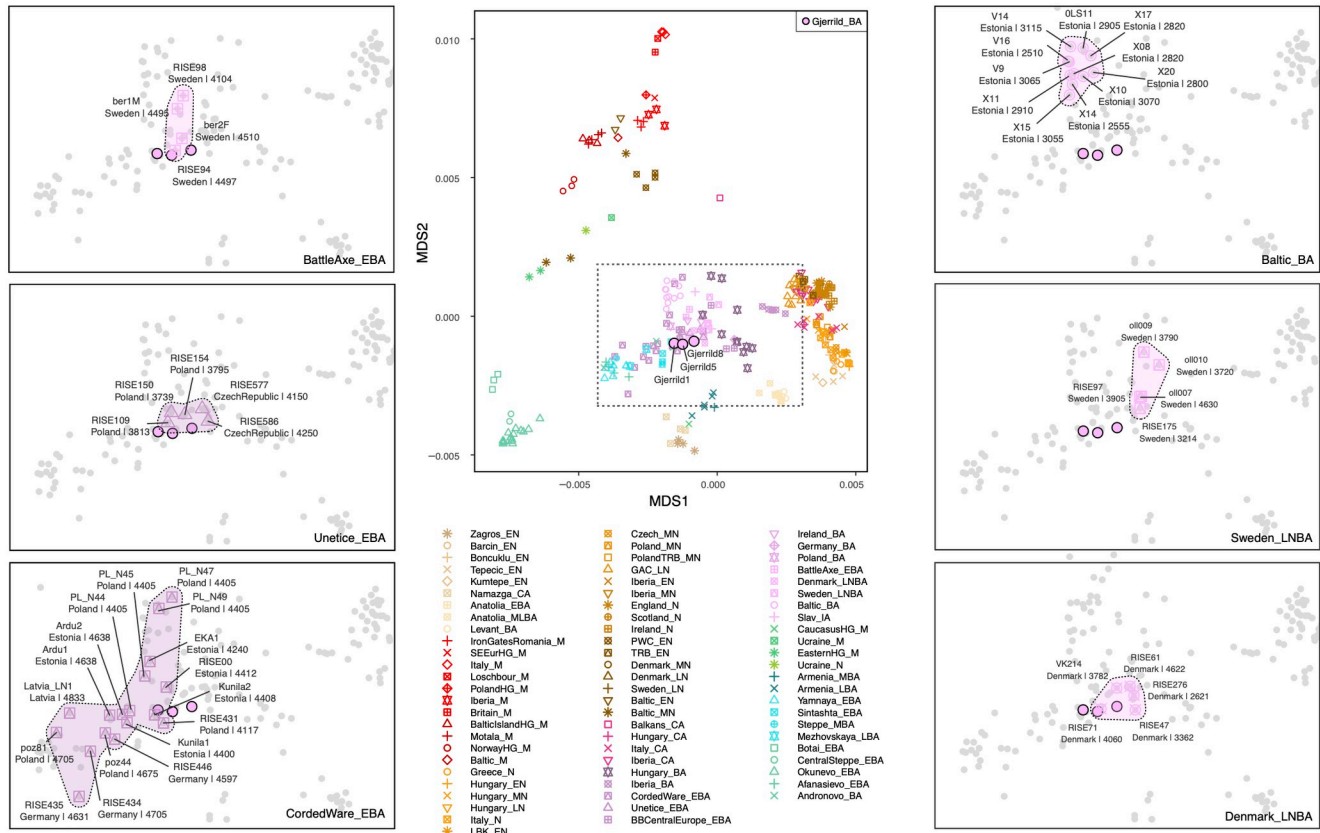

**Fig 4. Genetic clustering of the Gjerrild individuals.** Central panel shows Multi-dimensional scaling (MDS) plot of Gjerrild and 297 ancient individuals, based on genetic distances calculated from pairwise identity-by-state sharing of alleles. Side panels show enlarged view (dashed rectangle in center panel) highlighting the position of the three Gjerrild individuals in relation to various selected CWC, Late Neolithic, and Bronze Age groups. Individual labels in panels show country of origin and median calibrated $^{14}$C-age for each individual. Abbreviations: *M*, Mesolithic, *EN*, Early Neolithic; *MN*, Middle Neolithic; *LN*, Late Neolithic; *CA*, Chalcolithic; *EBA*, Early Bronze Age; *BA*, Bronze Age; *MLBA*, Middle-Late Bronze Age; IA, Iron Age, *HG*, Hunter-gatherer; *TRB*, Funnel Beaker Culture (Trichterbecher Culture); *SEEur*, South East Europe, *BB*, Bell Beaker; *GAC*, Globular Amphora Culture; *LBK*, Linear Pottery Culture. See S3 Table for complete overview of individuals (and references) included in these analyses.

Dimension two distinguishes the earlier European hunter-gatherer individuals as well as Neolithic individuals from Central Asia. It is evident that the Gjerrild individuals do not display the typical early/middle Neolithic ancestry which is observed among LBK, TRB and Globular Amphora Culture (GAC) groups. Rather, they genetically resemble individuals from Northern and Eastern Europe that post-date individuals connected with the Yamnaya migrations such as those related to the CWC, Battle Axe Culture and Unetice. On a more fine-scale level, they cluster somewhat separated from the earliest individuals connected with CWC, which are found genetically closer towards individuals from the Steppe. The ADMIXTURE results (Fig 5) confirms that all three Gjerrild individuals fall within the main "European Late Neolithic and Bronze Age" cluster, and correspondingly show Steppe-related ancestry components.

## Discussion

The SGC covered an area from the Netherlands to Jutland in Denmark where the Gjerrild grave is located. Our ancient DNA analyses of the Gjerrild skeletons have documented a close genetic relationship with individuals associated with the CWC observed as a steppe-related ancestry component which is not present in the previous Neolithic population linked to the

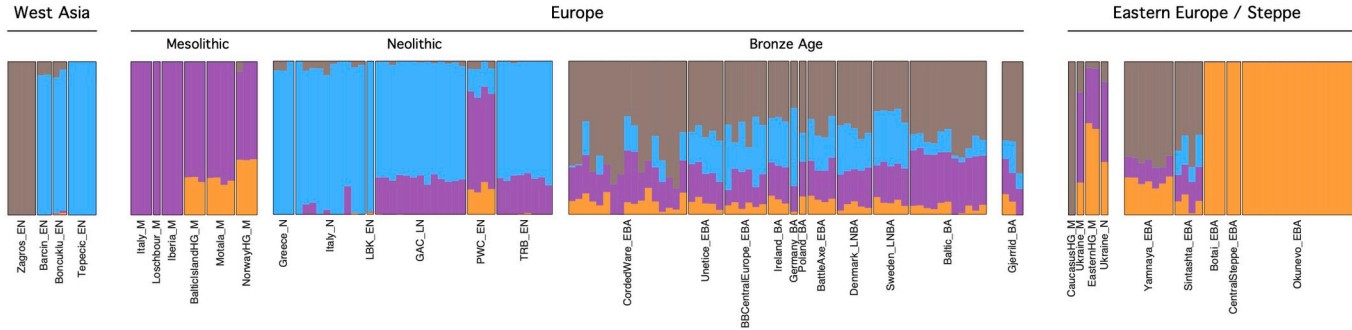

**Fig 5. Admixture analysis, selected groups.** Barplot of ancestry component proportions inferred using unsupervised ADMIXTURE, for K = 4 components. Gjerrild and other Late Neolithic / Bronze Age European individuals are characterized by a high proportion of a component associated with Steppe pastoralists (brown color, maximized in Iranian Neolithic individuals at this K value), but absent in preceding Neolithic farmer groups. See S3 Table for complete overview of individuals (and references) included in these analyses.

TRB Culture. Our results thus suggest that the population that lived and buried their dead according to SGC customs should be considered genetically a northwestern branch of populations connected to the CWC. The SGC appeared from around 2850–2700 BCE in central and western Jutland, whose flat landscape with sandy soils was sparsely populated by preceding Neolithic groups [17]: Fig 5.5. Here forests were also rather open, and could more easily be transformed into pastures, as evidenced in pollen diagrams, a process already under way in areas settled by late TRB groups pursuing a pastoral economic strategy in this area [9, 63, 64]. Based on archaeological indicators, the origin of the SGC migration could point towards central Germany, the Halle region [13], and it must have been substantial and continuous, as the forests disappeared within a few generations. Once the SGC had established their territory in central and western Jutland it seems likely that they expanded into areas possibly still populated by late TRB groups in northern and eastern Jutland, as well as the Danish islands [16]. The individuals from the Gjerrild grave are from this secondary phase of expansion.

## Age and use of the Gjerrild barrow

All burials in the Gjerrild Barrow seem to be primary interments of complete bodies, although a thorough modern examination of the total bone material is still lacking. It can be noted that the documented body positions were variable, as individuals dated to the SGC period were found both in the expected hocker position (Gjerrild 1) and in extended supine positions (Gjerrild 6 and 7). Our new radiocarbon dates suggest a main period of use from ca 2600/2500 cal BCE cal to probably after 2200 cal BCE, with reuse at c. 1700 cal BCE (S1 Fig). These dates indicate a longer period of use than was inferred from the artefacts, and the construction of the grave can now be set to the middle part of the SGC period (the Ground Grave period). This is based on the dating of Gjerrild 8, the oldest individual buried. The remaining dates range across the SGC period, with Gjerrild 5 being the youngest. Depending on the definition of the regional MN/LN period boundary in southern Scandinavia, this individual can be classified as either Late Neolithic or bordering the MN/LN transition. It is evident that the individuals were not buried simultaneously but placed in the chamber over a period of time, roughly estimated to c. 300 years. If no more than 10 people were interred, on average only one person per generation could have been buried, which implies a strong selection from the total population. Whether the individuals placed in the chamber belonged to the same family lineage cannot be determined from these results but it is clear that the selection does not follow sex or age criteria.

As noted above, the Gjerrild 6 skeleton had a flint arrowhead of the so-called D type embedded in the breast bone. Arrowheads of this type are thought to belong to the later SGC (c. 2525–2250 cal BCE, [6]: 438–9) and this is confirmed by our radiocarbon date of the Gjerrild 6 skeleton (Table 1). The excavator of the monument suggested that Gjerrild 6 was the last person to be buried in the grave but this is now contradicted by the $^{14}$C dates [22]. Type D arrowheads derive from the classical PWC tanged arrowheads (types A-C) as shown by morphological similarities and hybrid (C-D) forms. However, their relationship is also reflected in the eastern distribution of type D arrowheads, including a concentration of finds on Djursland, which overlaps with the former PWC areas. Type D arrowheads are short (often 5–7 cm), three-sided, completely chipped points that appear technically poorer than type C, which they derive from. Both type C (PWC) and type D (SGC) are probably to be regarded as specialized war arrowheads, which is confirmed by the Gjerrild find [64, 65].

The largest of the two pottery vessels [22]: Fig 5 left, [23]: Fig 31, was found approx. in the middle of the stone cist. Its height is 16.7 cm and its width 20.5 cm. It is a fine and typical example of the round-bodied type $I_1$ [18] the distribution of which centres in Himmerland, the area to the north of eastern Jutland. The other, smaller pottery vessel lying close to the skull of the male individual 6 in the southern end of the grave is 9.5 cm high and has a rim diameter of 11.7 cm. It belongs to a large group of straight-walled beakers but has incurved sides [22]: Fig 5 right, [23]: Fig 31, such as type $L_3$ [18]. Examples of this type are found in the same area of north-eastern Jutland, while one is actually from western Djursland. Both vessels are dated to the late SGC, c. 2450–2250 BCE [6], and both of them connect at least some of the buried individuals, if not all, to the inland cultural environment of the late SGC, as does the grave type.

A thin-bladed, thick-butted flint adze with a slightly hollowed edge was found in the southern part of the grave near individual 6 and had a length of 13.1 cm and edge-width of 5.9 cm. The flint adzes began to appear during the Ground Grave Period of the SGC [18]. The thin-bladed adze from the Gjerrild grave has been classified as type 3A1, which is dated to the Upper Grave Period of the late SGC [6]: Fig 259c, 370, thus corresponding to the date of the pottery vessels. The flint adze from the grave may answer why the area around Gjerrild was targeted by people of the late SGC. This kind of flint adze has a pronounced distribution in eastern Djursland [66]: Fig 9. At the neighboring Gjerrild Klint, abundant flint resources of high quality were located [67]. Control of such flint resources would have given these newly established communities an advantage in producing and exchanging higher quantities of different flint artefacts, thus creating larger network alliances beyond the local region.

## Diet and mobility

Collagen $\delta^{13}$C and $\delta^{15}$N isotopes in bone carry information on the diet of the individual, mainly concerning the sources of the protein component. In environments such as Denmark, where vegetation is based on the C3 photosynthetic pathway, the main use of $\delta^{13}$C values is to estimate the amount of marine versus terrestrial protein in the diet [68]. In a Danish Neolithic context, a completely terrestrial diet would be expected to give $\delta^{13}$C collagen values around -22 ‰ while a completely marine diet would lead to values at ca -10‰ [69]. In contrast, $\delta^{15}$N values depend on the trophic level of the protein sources, with a stepwise enrichment of ca 3–5‰, although this varies between species. The enrichment for humans may be even higher and has been estimated to ca 6‰ [70]. In northern European contexts, herbivores normally range ca 2–6‰ and carnivores ca 5–10‰. Due to the longer food chains in marine and freshwater systems, $\delta^{15}$N values in top predators can be substantially higher than in terrestrial systems. The detailed interpretation of human diet depends on mapping baseline values throughout the

food chain. Such values may vary both geographically and temporally due to variables related to climate, vegetation cover, or human practices such as manuring. In the absence of local and chronologically relevant background knowledge, interpretation of the Gjerrild isotope values can only be rather general. Most of the $\delta^{13}$C values range from ca -19 to -19.5‰ (Table 1, Fig 2) and these are values consistent with a limited intake of marine proteins, despite the location close to the coast. We estimate that the proportion of marine protein could be around 15–20% of the protein intake. The adult female, individual 1, deviates by having a completely terrestrial $\delta^{13}$C signal. Examining the δ15N content, we observe that the adults have quite low values, ranging from 8.3 to 9.5‰. The two children have higher values (11.5 and 12.4‰), most likely due to a remaining lactation effect. The values of the adults are similar to or slightly lower than, those measured in other Neolithic contexts in northern Europe [69, 71, 72] and clearly lower than values from Corded Ware groups in Germany [73]. With the caveat that the isotopic baselines applied (from [69]) do not represent the specific geographical area and period, these values suggest only a moderate intake of protein from animal sources, and they cannot be distinguished from earlier and later populations in Denmark (Fig 2). These values also argue against any significant amount of marine or freshwater protein intake, leaving plants (i.e. cereals) as a likely major contributor. The dietary isotope results for the Gjerrild individuals corroborate settlement and economic data that have indicated a higher reliance on agriculture and a less mobile lifestyle in the later part of the SGC than during its first centuries [6, 74].

87Sr/86Sr ratios have previously been measured for all the sampled individuals and are presented in a comprehensive study by Frei et al. [32]. A more detailed mapping of Sr isotope variation on the Djursland peninsula is now available [75]. Based on this, a slight revision of the previously published interpretation seems appropriate.

The disarticulated Bronze Age mandible (RISE72) yielded the lowest 87Sr/86Sr value, consistent with the immediate surroundings. Four individuals, Gjerrild 5, 6, 7, and 8, showed values 0.710–0.711. These values are higher than what is found in the immediate surroundings. The two adult males, Gjerrild 6 and 7, showed very similar values and may have been growing up in the same area. The nearest area with such values is found c. 10–20 km to the west. Similar values can also be found in other Danish regions, as well as in southernmost Sweden and many areas of central Europe. The question of their origin can therefore not be answered conclusively. However, the simplest explanation for the Sr isotope variation shown by these individuals is that the uptake area for burial in the Gjerrild grave encompassed areas at least some 10–20 km distant from the site. The adult female, Gjerrild 1, showed the highest Sr isotope ratio of all, 0.712764 [32]. This value is clearly outside the baseline range of Denmark, as presently known, and suggests long range migration for this individual. As noted above, she also had a different diet with no marine component. Possible areas of origin include the southern parts of Sweden, Bornholm, and several regions in central Europe, but it is not possible to distinguish between these from either isotopic or archaeological data. In any case, the diverging Sr signature of this female individual is compatible with an occurrence of exogamic practices, as also suggested for CWC groups in southern Germany [73].

## Genetic ancestry

As often observed for ancient skeletal remains from temperate regions, the DNA preservation of the Gjerrild samples was poor except for the petrous bone available from Gjerrild 5, yielding a relatively high endogenous DNA content. This result confirmed that petrous bone is the favourable choice for aDNA analysis, in particular when preservation is problematic (e.g. [57]). Three individuals yielded sufficient genome-wide data to be included in the analyses, peaking with >1X genomic coverage in Gjerrild 5.

Our clustering and admixture analyses clearly demonstrate that the three Gjerrild individuals carry a significant proportion of a specific genomic ancestry component that is often referred to as "steppe DNA". This is the genomic signature of the Yamnaya migrations around 3000 BCE that forever transformed the genetic landscape of Europe [1, 2]. Prior to 3000 BCE the Neolithic farmers of Europe displayed genomic ancestries tracing directly back to large scale population expansions from the Middle- and Near East starting around 8000 BCE. However, around 3000 BCE the nomadic Yamnaya herders from the Pontic Caspian Steppe expanded both eastwards to the Altai Mountains and westwards into Europe. This meeting with the local Neolithic communities may sometimes have been dramatic and violent but created a cultural and genetic melting pot resulting in the formation of a new cultural horizon, namely the CWC. Therefore, when ancient DNA from skeletons of the CWC and related cultures is being analysed, a mix of European Neolithic and Yamnaya-related ancestry is always revealed [1, 2].

The ancestry analyses of the Gjerrild skeletons show that the SGC in Denmark could be characterised genetically as a northern 'branch' of CWC. As such, the transition from the TRB to SGC is very unlikely to be characterized as simple demographic continuity. Rather, the appearance of SGC is a product of the same major demographic transition that was taking place in most parts of Europe around 3000 BCE. Given our current knowledge on this continent-wide demographic transition, this is not a surprising result. However, our observation still represents an important piece of information towards understanding the geographic extent and timing of the CWC expansions, and it thus contributes to a long-lasting debate on the origin and later expansion of the SGC in Denmark. More specifically, the three individuals represent a later phase of a local eastward migration towards eastern Jutland, whose archaeological and ecological context we shall discuss in more detail in the next section.

Because we only have genetic data from three individuals, of which only one (Gjerrild 5) has >1X genomic coverage, we will not engage in a detailed genetic discussion of the exact origin of the Danish SGC and relationship with other CWC-like groups. However, it does seem that the Gjerrild individuals cluster somewhat separate from the main European CWC cluster, who on average display a slightly higher proportion of Steppe ancestry (Figs 4 and 5). The three Gjerrild individuals carry close genetic resemblance to individuals from other CWC-derived cultures such as Swedish Battle Axe individuals, Unetice individuals from Poland, and five previously-analysed late Neolithic/early Bronze Individuals from Denmark (Fig 4). These observations hint at subtle geographic or temporal population genetic structuring in the broader CWC horizon. It suggests that the genetic composition of the Gjerrild individuals resulted from long-term processes of interaction with still existing Neolithic populations but this requires more in-depth analyses of larger population genomic datasets to be tested properly.

Unfortunately, we do not have genetic data available from Danish PWC individuals nor from the PWC in western Sweden. However, we can firmly establish that the Gjerrild individuals show no genetic affiliation with the PWC individuals known from eastern Sweden [76], as these do not display steppe ancestry (Figs 4 and 5). PWC groups were highly mobile, and we see PWC sites in northern Jutland and Djursland from c. 3000 BCE continuing into the early SGC phase. So, if the eastern Swedish PWC individuals can be regarded as genetic representatives of this culture in broader terms due to their mobility, the PWC-SGC interaction during the early phase of the SGC would be one of two highly distinct population groups, similar to the broader CWC-TRB interaction occurring throughout large parts of Europe at this point in time. However, this proposition needs to be tested by future DNA analyses of individuals connected to the western PWC.

The mtDNA haplogroups of Gjerrild 8 and 5 were both determined to be from the K2a lineage whereas Gjerrild 1 was HV0. These haplogroups are part of the "Neolithic package" that became common in Europe following the Neolithic transition [77] and the K2 subgroup has previously been observed both among subsequent CWC [78] and Bell Beakers from England [79]. This has often been interpreted as continuity in the female gene pool, suggesting that the incoming CWC-related migrants were largely males and then local females were bought in [3]. Our mtDNA results from the Gjerrild grave are not inconsistent with this notion. More unusual is the Y-chromosome haplogroup of Gjerrild 5 determined to R1b-V1636 (R1b1a2) (Fig 3). This rare subclade of R1b has previously been observed in four Eneolithic individuals from the Pontic-Caspian Steppe [2, 58] as well as a Chalcolithic individual from Anatolia (Arslantepe) [59] R1b1a2 among [59].

## Archaeological implications and interpretations

The timing of the SGC arrival in Denmark can be corroborated by pollen analyses. Of particular significance for understanding the Gjerrild site is the fact that PWC activity on northern Djursland ended around the 28[th] century BCE, followed by an almost complete hiatus in archaeological finds in the area [21, 80]. This hiatus came to an end around the time of the earliest Gjerrild burial, which we have now dated accurately to the 26[th] century BCE. In a high-resolution pollen diagram from Fuglsø bog in northern Djursland, some 12 kilometres west of Gjerrild, we can observe this 200-year hiatus in farming activity, allowing the forest to regenerate [81]. When SGC arrived in Northern Djursland, the forest was once again opened, now mainly for permanent grazing. An increase in dust dispersal can also be observed as indicative of open fields, but less pronounced than during the earlier TRB farming regime. The Gjerrild barrow thus marks a time of renewed settlement in the area after approximately two centuries of 'dark ages' [21:482].

Since human remains are sparse from the SGC, the DNA data from the Gjerrild individuals have provided a unique insight on the process underlying the cultural and social change connected with SGC in Denmark during the third millennium BCE. The present study suggests that the regional SGC in Denmark not only represented a cultural transformation but also a genetic change connected to the spread of CWC in other parts of Europe. How this transformation was perceived locally is not clear. However, if we look at the sparse information available, it indicates that encounters with PWC groups were not always peaceful. The PWC in Denmark is characterized by relatively few larger settlement sites primarily found in the coastal areas of northern and northeastern Jutland, including the significant presence in northern Djursland. The economy seems to have been mixed, focused on marine resources but including game, livestock and small-scale farming [19, 82, 83]. Initially, PWC and SGC communities probably did not compete seriously for territories and resources. However, this would likely have changed as the SGC expanded to the north and northeast as indicated by four SGC graves from the Under Grave period (c. 2850–2600 cal BCE) containing specialized PWC war arrowheads (type C). The location of the arrowheads within the graves makes it plausible that they were logged in the bodies at the time of burial (see [19], catalogue and [16] for further details and references). The four single graves are all found in what can be defined as the SGC/PWC border zone in northern Jutland just south of the Limfjord. Three of the graves are among the most northerly recorded early SGC graves placed in what might be considered PWC core territory. Thus, we must expect violent encounters to have taken place in the form of raids or short-lived feuds in areas of interest to expanding SGC groups as well as the existing PWC population. Such a pattern would indeed align well with evidence from other regions in

Europe indicating that violent conflict played a significant part in the turbulent demographic and cultural processes of the 3$^{rd}$ millennium BCE [84, 85].

The presence of type D-arrowheads and the relatively late date of the Gjerrild burial makes it unlikely that the traumas found on some of the individuals are the results of hostile PWC-SGC encounters. Instead, they are more likely to be the outcome of intra-SGC competition. An obvious source of conflict could be the rich flint deposits found at the Gjerrild coastal cliff less than 1.5 kilometers east of the stone cist barrow analyzed here. These resources were intensively exploited during the PWC period [67], and probably also during the subsequent SGC epoch as indicated by certain concentrations of thick-butted flint adzes and type D arrowheads ([21], 482 with references). In any case, it is noteworthy that the two individuals associated with arrowheads and traumas (Gjerrild 6 and 7) have overlapping dates and might be contemporary. This opens for a possible scenario where both males could be victims of the same or perhaps a related series of hostile encounters between SGC groups settling in former PWC areas. As noted above, Gjerrild and the few contemporary SGC sites on northern Djursland represent a resettlement after approximately two centuries during which the area appears to have been abandoned, potentially allowing for a more unrestricted or communal access to the high-quality flint deposits which could be accessed easily from the sea. When the group that constructed the Gjerrild barrow settled in the area during the 26th century BCE, it may thus have been one of several groups staking a territorial claim to an area and to resources that had for a long time been contested or unregulated. This scenario could provide a potential explanation for the violent conflicts reflected in the Gjerrild skeletons.

## Supporting information

**S1 Fig. Radiocarbon dates.** Calibration plot of the radiocarbon dates from Gjerrild, based on reservoir corrected dates. Individual 7 was dated twice as part of a quality control procedure at the Oxford laboratory. A combined date was calculated for this individual. Horizontal bars show 95.4% probability ranges.
(TIF)

**S2 Fig. PCA plot.** Genetic relationship of the three Gjerrild and other ancient individuals based on PCA. The ancient genomes were projected onto the modern variation of the Affymetrix Human Origins panel.
(TIF)

**S1 Table. Sample information.**
(XLSX)

**S2 Table. Sequencing statistics.**
(XLSX)

**S3 Table. Ancient comparative datasets analyzed in this study.**
(XLSX)

**S4 Table. Counts of haplogroup-defining SNPs in Gjerrild 5 for selected R1b-subhaplogroups.**
(XLSX)

## Acknowledgments

We thank Jesper Stenderup and the staff at the Danish National High-throughput Sequencing Centre for technical assistance.

## Author Contributions

**Conceptualization:** Kristian Kristiansen, Morten E. Allentoft.

**Data curation:** Karl-Göran Sjögren, Niels N. Johannsen, Poul Otto Nielsen, Lasse Sørensen, Rune Iversen.

**Formal analysis:** Anne Friis-Holm Egfjord, Ashot Margaryan, Karl-Göran Sjögren, T. Douglas Price, Martin Sikora, Morten E. Allentoft.

**Funding acquisition:** Eske Willerslev, Kristian Kristiansen, Morten E. Allentoft.

**Investigation:** Anders Fischer.

**Project administration:** Anne Friis-Holm Egfjord, Kristian Kristiansen.

**Supervision:** Ashot Margaryan, Morten E. Allentoft.

**Validation:** Karl-Göran Sjögren, Rune Iversen.

**Writing – original draft:** Anne Friis-Holm Egfjord, Kristian Kristiansen, Morten E. Allentoft.

**Writing – review & editing:** Ashot Margaryan, Anders Fischer, Karl-Göran Sjögren, T. Douglas Price, Niels N. Johannsen, Poul Otto Nielsen, Lasse Sørensen, Eske Willerslev, Rune Iversen, Martin Sikora, Kristian Kristiansen, Morten E. Allentoft.

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
