## [Decision Letter · Decision Letter 0]

15 Oct 2020

PONE-D-20-24197

GENOMIC STEPPE ANCESTRY IN SKELETONS FROM THE NEOLITHIC SINGLE GRAVE CULTURE IN DENMARK

PLOS ONE

Dear Dr. Margaryan,

Thank you for submitting your manuscript to PLOS ONE. After careful consideration, we feel that it has merit but does not fully meet PLOS ONE’s publication criteria as it currently stands. Therefore, we invite you to submit a revised version of the manuscript that addresses the points raised during the review process.

Please address all comments before re-submission. In particular, elaborate on the definition of 'archaeological cultures' vs. 'biological populations' as well as on the sample size and sample quality. 

We look forward to receiving your revised manuscript.

Kind regards,

Peter F. Biehl, PhD

Academic Editor

PLOS ONE

Additional Editor Comments:

Your manuscript has now been seen by two referees, whose comments are appended below. You will see from these comments that the referees find your work of great interest, but they have raised several issues that must be fully addressed before re-submission.

Journal Requirements:

2. In your Methods section, please provide additional information regarding the permits you obtained for the work. Please ensure you have included the full name of the authority that approved the field or museum site access and, if no permits were required, a brief statement explaining why.

3. In your manuscript, please provide additional information regarding the specimens used in your study. Ensure that you have reported specimen numbers and complete repository information, including museum name and geographic location. If permits were required, please ensure that you have provided details for all permits that were obtained, including the full name of the issuing authority, and add the following statement:'All necessary permits were obtained for the described study, which complied with all relevant regulations.'If no permits were required, please include the following statement:'No permits were required for the described study, which complied with all relevant regulations.'For more information on PLOS ONE's requirements for paleontology and archaeology research, see https://journals.plos.org/plosone/s/submission-guidelines#loc-paleontology-and-archaeology-research.

"This research in this study was part of the

project “Towards a new European Prehistory” funded by the Riksbanken Jubileumsfond to

KK. Centre for GeoGenetics is funded by the Lundbeck Foundation. MEA is supported by

the Independent Research Fund Denmark."

" KK received funding by the Swedish Riksbanken (The Swedish Foundation for Humanities and Social Sciences) grant M16-0455:1 - Towards a New Prehistory.

https://www.riksbank.se/en-gb/

Reviewers' comments:

Reviewer's Responses to Questions

**Comments to the Author**

1. Is the manuscript technically sound, and do the data support the conclusions?

Reviewer #1: Yes

Reviewer #2: Yes

2. Has the statistical analysis been performed appropriately and rigorously? 

Reviewer #1: I Don't Know

Reviewer #2: Yes

3. Have the authors made all data underlying the findings in their manuscript fully available?

Reviewer #1: Yes

Reviewer #2: Yes

4. Is the manuscript presented in an intelligible fashion and written in standard English?

Reviewer #1: Yes

Reviewer #2: Yes

5. Review Comments to the Author

Reviewer #1: The paper presents new and significant aDNA analyses for three Neolithic individuals from Jutland and a discussion of their archaeological and genetic contexts. As Jutland is one of the Core Areas of the European Corded Ware, which features key in the current archaeogenetic debates, but is characterized by extremely poor bone preservation, these results are the first of their kind, and thus of high relevance for an international audience and should definitely be published.

It seems to me that the molecular biological investigations (whose details I cannot really assess, due to my lack knowledge). The interpretation of the genetic results in the context of our current archaeogenetic knowledge need some, rather minor, revisions, the way in which genetic and archaeological informations are combined is, in my view at times problematic, however, it is possible to resolve these problems by minor revisions. The archaeological analyses and the final presentation of a migration narrative are in my view a little speculative, but I would argue that they are totally acceptable.

Overall, my recommendation is thus to accept the paper with minor revisions.

Those revisions I would like to recommend are the following:

When it comes to the comparison of archaeological and genetic data there are terminological issues, which really muddy the water, and which should, in my view, be resolved, because they also lead to tautological statements and faulty conclusions. They all derive from the widespread fallacy to treat archaeological cultures as if they would represent specific social groups of humans, and distinct biological populations. This really constitutes a big problem in archaeology. And it is a problem that has plagued the archaeogenetic project from its beginning. It is really time we get past it. The fallacy of treating archaeological cultures like human social groups becomes visible already in the abstract, where it is stated that “

“Assuming that the Gjerrild skeletons are genetically representative of the SGC in broader terms, this culture cannot be regarded as a demographic continuation of the previous Neolithic population.”

It is more than a pedantic archaeological trivialty: A ‘culture’ in the archaeological sense is a group of archaeological finds, created by archeologists, a population is a unit of biological organisms, defined by biologists. To conflate the two, as it is done in this sentence or to simply equate them has created a lot of misunderstandings. The conflation is the main justification for the assumption at the start: namely that the genetic profiles of three individuals from one single burial could ever be representative for the genetic profiles of all other individuals buried in the context of similar burial types (thus connected to the same archaeological culture). This assumption is a very strong assumption and does, as it stands seem to indicate that all individuals connected to SGC burials would have the same genetic profile. There are actually real, but totally different, arguments to assume that it is in fact correct, but this does not come from the Danish SGC, but rather from the prior knowledge that the great majority (virtually all) individuals connected to Corded Ware Single Graves (a variant of which the SGC burials are) actually have high amounts of steppe ancestry, while all individuals connected to TRB do not. But this cannot be taken fro granted and it does not follow from the analyses of the three Gjerrild individuals alone.

Also “this culture cannot be regarded as a demographic continuation” should be replaced by something like: „the transition from the TRB to SGC is very unlikely to be connected to demographic continuity.”

This kind of terminology continues on the following pages, where “groups of the Yamnaya culture expand (p. 3)” (I would change that to "groups connected to Yamnaya" – an easy way to get rid of the problematic associations), or the statement that the Pitted Ware Culture (PWC) would have a specific economy (p. 4). Or: “Rather, they genetically resemble individuals from Northern and Eastern Europe that post-date the Yamnaya migrations such as the CWC, Battle Axe Culture and Unetice (p.16).” This can easily be resolved by writing: … such as individuals connected to the CWC, Battle Axe Culture and Unetice.

That there is more to this than terminological subtleties is illustrated by the fact that it leads exactly to a very simplified research question being supposedly answered, namely a binary "continuity vs. discontinuity of populations":

p. 17: “…and we can therefore reject that SGC in Denmark represents a demographic continuation of earlier Neolithic TRB groups".

It is unfortunately that the whole discussion is broken down to such a simplistic "continuity or not"- question, which only has any relevance if one presupposes that those units are clear-cut bounded groups of people. This is not to question that migration is obviously an important factor resulting in the shaping of the SGC, but simply to say yes or now to the migration question is really not doing the subject any favour. Especially as later in the archaeological part, continuity of female lineages is basically admitted. So there is a much more interesting story here than the mere identification of “migration as opposed to continuity”. This is then also acknowledged in pages 22-23:

“Therefore, when ancient DNA from skeletons of the CWC and related cultures is being analysed, a mix of European Neolithic and Yamnaya-related ancestry is always revealed [1,2].

The ancestry analyses of the Gjerrild skeletons show that the SGC in Denmark could be characterised genetically as a northern ‘branch’ of CWC. As such, the SGC is not a simple demographic continuation of the previous Neolithic TRB people”

Why not use this kind of argument already on page 17?

The following sentence (on page 17) is also problematic:

“Rather, our results suggest that the population that lived and buried their dead according to SGC customs should be considered a northwestern branch of the CWC – at least in genetic terms”.

This could be reformulated, avoiding the impression that CWC would be something which can be expressed in genetic terms. This is even easier as archaeologically, CWC and SGC are 80-90% identical anyway.

On page 23 it is stated:

“However, it does seem that the Gjerrild individuals cluster somewhat separate from the main European CWC cluster”

This is maybe the place where it also should be acknowledged and discussed that the Gjerrild individuals also do not derive from a typical SGC nor a typical CWC context, which is relevant in this whole discussion. But again I would urge to be careful not to conflate biology and culture.

Pages 23-24:

“So, if the eastern Swedish PWC individuals can be regarded as genetic representatives of this culture in broader terms, the PWC-SGC interaction occurring on Djursland would be one of two highly distinct population groups, similar to the broader CWC-TRB interaction occurring throughout large parts of Europe at this point in time.

This sentence again illustrates the dangers of the conflation of archaeological cultures and biological populations: a tautology: assuming that individuals connected to Danish PWC would be genetically identical to Swedish PWC (for which there is no real argument), then PWC in Denmark and SGC in Denmark would be different. With other words: assuming they were different, we can conclude they are different…

P. 24:

“This rare subclade of R1b has previously been observed in three Eneolithic individuals from the Pontic-Caspian Steppe [2,76] but it pre-dates the expansion and diversification in the R1b-M269 lineage, which is typically considered to be associated with the Yamnaya migration and thus common across present-day Europeans. As such the genetic profile of Gjerrild 5 broadens our known Ychromosomal evidence for the links to the early steppe societies.”

This seems like a fact that merits more discussion. Does this mean that the Y-chromosome haplogroup of the Gjerrild individual is not among the ones known from Yamnaya burials? I guess this needs clarification, as the Yamnaya – relation of steppe ancestry is featuring so prominent in the paper. Also, it has become increasingly clear that individuals buried in Yamnaya burials analysed so far are definitely not the biological ancestors of individuals buried in early Corded Ware graves. I would argue that it is in general not really a good idea anymore to narrow down Steppe ancestry to individuals from Yamnaya burials.

And a few really small things:

P.3 “The appearance of the Single Grave Culture (SGC) in Denmark, Northern Germany and Netherlands c. 4850-4600 years ago during the Nordic Middle Neolithic period (MNB)”

They do have different terminologies in Northern Germany (Younger Neolithic), and in the Netherlands (Late Neolithic).

P.6 “Hans Christian Petersen carried out a comparative osteological analysis of two Gjerrild craniums that were compared to skulls of the previous TRB, as well as to those of CWC individuals from central Germany [26].”

This is a rather uncritical reference to this study. How can something seriously be said to “group with” anything at a sample size of two?

P. 17 “The SGC covered an area from the Netherlands to Jutland in Denmark where the Gjerrild grave is located.”

The Dutch and the Danish SGCs are not so similar (not more similar at least than Danish SGC and other CWC regions). It is more a coincidence of resaearch history that they ended up having the same name. I would recommend in general to subsume SGC and Battle Axe Culture as variants of CWC, but of course there are different conventions here.

Overall, I hope my critical tone in the discussion of individual sentences and statements does not obscure the fact that the study is overall very valuable and has a lot of interesting, relevant insights and arguments. Especially the discussion of migration scenarios into Djursland is an interesting and and productive way of thinking about contextualization of the archaeogenetic findings. Resolving the terminological issues I addressed will, however, make this a much stronger paper. Although I do suggest several revisions, they are all, really, minor.

Reviewer #2: Broad Comment

• I really enjoyed this manuscript. Personally, seeing a genetics paper that clearly displays an encyclopedic knowledge of the archaeological context of the finds is both impressive and encouraging. Unfortunately, I feel that some of the findings may be over-ambitious based on the (a) very small sample size, and (b) quality of the samples. I in no way mean this disrespectfully, as finds in this region, and certainly of this age, are rare, and great care and cost was made to produce this data. However, I'd like to see more care taken with the depth of finality to the findings, in light of these limitations.

• I'd really like to see more made of your interpretation of the R1b-V1636 sample in context with other information. The mtDNA HG and the YHG are both Neolithic, and specifically the YHG is rare. However, this individual also carries step ancestry, so how does this compare to the other R1b-V1636 samples? Is this cocktail of history common, to be expected, does it shed any light on the possibility of a "Neolithic ancestry refuge" that was afforded to this population due to their northerly isolation?

Page 4:

• “facilitated by migration of people” could be “facilitated by a/the migration of people” or “facilitated by migrations of people”.

Page 5:

• “well-preserved” seems a little optimistic. Perhaps “relatively well-preserved” better reflects the high quality of the samples, given the typical soil conditions?

• “With this follows…” could be “From this follows…”.

Page 6:

• “late in date” is ambiguous. Perhaps “more recent in date”?

Page 15:

• R1b-V1636 is indeed an interesting haplogroup. Do you state whether it is definitely no more diverged, i.e., ancestral for R1b1a2a and R1b1a2b?

• You mention SNP counts for each sample, and percentages, but it might be worth restating that you’re using the 12,731,663 transversions from the 1000GP for clarity.

Page 16:

• Why has PCA not been used here? I’m not suggesting that MDS is not a valid method, but many researchers in this field would likely appreciate a familiar comparison between this research and many other papers by including a classical PCA of West Eurasian modern populations (say that from Haak2015). If the reason is poor coverage on the 1240K SNPs (which I imagine there would be), then this could be stated.

• Figure 5 the labels have issues in spacing.

• “Well-established clustering of the ancient individuals related to age, geography and cultural contexts” is ambiguous for the word age. I assume you mean the calibrated sampling age, and not in fact the age at death of the individual.

Page 17:

• I find the language “and we can therefore reject that SGC in Denmark represents a demographic continuation of earlier Neolithic TRB groups” quite strong language based on one relatively high coverage sample, and two low coverage samples. Specifically, the finding of an individual carrying a rare pre-Steppe R1b Y-haplogroup that also carries Steppe ancestry in the literature, and to assume that this individual (plus two lower-coverage individuals) represents a Denmark-wide culture is perhaps over-reaching slightly based on a small number of (albeit very impressive and rare) samples.

Page 22:

• “expanded both eastwards to Altai Mountains and…” should be “expanded both eastwards to the Altai Mountains and…”.

Page 23:

• Again, using the language “settles a long-lasting debate on the origin of the SGC” is particularly strong, based on the evidence of just three skeletal samples from the same location (even if they come from a 20km-wide radius). I concede that in light of contemporary, and geographically nearby populations (such as northern German), that these findings likely will stand up to (hopefully) further finds, however, I still find the finality of the language a little too strong. The authors do state that due to the limitations of the sample numbers and quality, that finding the exact origin of the Danish SGC is unrealistic, although it’s actually likely impossible at this stage. I’d like to see this reservation extended to other facets of the interpretations.

Page 24:

• Another Eneolithic R1b-V1636 individual was also found in Arlanstepe, Turkey [Skourtanioti2020].

6. PLOS authors have the option to publish the peer review history of their article (what does this mean?). If published, this will include your full peer review and any attached files.

Reviewer #1: No

Reviewer #2: No

---

## [Author Response · Author response to Decision Letter 0]

17 Dec 2020

We have included all our replies to the editor's and reviewers' comments in the Gjerrild_ResponseToReviewers.docx.

---

## [Editor Report · Decision Letter 1]

18 Dec 2020

Genomic Steppe ancestry in skeletons from the Neolithic Single Grave Culture in Denmark

PONE-D-20-24197R1

Dear Dr. Margaryan,

We’re pleased to inform you that your manuscript has been judged scientifically suitable for publication and will be formally accepted for publication once it meets all outstanding technical requirements.

Kind regards,

Peter F. Biehl, PhD

Academic Editor

PLOS ONE
---

## [Editor Report · Acceptance letter]

6 Jan 2021

PONE-D-20-24197R1 

Genomic Steppe ancestry in skeletons from the Neolithic Single Grave Culture in Denmark 

Dear Dr. Margaryan:

I'm pleased to inform you that your manuscript has been deemed suitable for publication in PLOS ONE. Congratulations! Your manuscript is now with our production department. 

Kind regards, 

on behalf of

Dr. Peter F. Biehl 

Academic Editor

PLOS ONE